# Microhomology Selection for Microhomology Mediated End Joining in *Saccharomyces cerevisiae*

**DOI:** 10.3390/genes10040284

**Published:** 2019-04-08

**Authors:** Kihoon Lee, Jae-Hoon Ji, Kihoon Yoon, Jun Che, Ja-Hwan Seol, Sang Eun Lee, Eun Yong Shim

**Affiliations:** 1Department of Molecular Medicine, Institute of Biotechnology, University of Texas Health Science Center at San Antonio, 7703 Floyd Curl Drive, San Antonio, TX 78229-3900, USA; y2k0108@gmail.com (K.L.); seolj@uthscsa.edu (J.-H.S.); 2Genomic Instability Research Center, Ajou University School of Medicine, 164, World Cup-ro, Yeongtong-gu, Suwon 16499, Korea; jij@ajou.ac.kr; 3Department of Radiation Oncology, University of Texas Health Science Center at San Antonio, 7703 Floyd Curl Drive, San Antonio, TX 78229-3900, USA; yoon.kihoon@hotmail.com (K.Y.); chej@uthscsa.edu (J.C.)

**Keywords:** microhomology, microhomology-mediated end joining, DNA double strand break, mismatch, deletion

## Abstract

Microhomology-mediated end joining (MMEJ) anneals short, imperfect microhomologies flanking DNA breaks, producing repair products with deletions in a Ku- and *RAD52*-independent fashion. Puzzlingly, MMEJ preferentially selects certain microhomologies over others, even when multiple microhomologies are available. To define rules and parameters for microhomology selection, we altered the length, the position, and the level of mismatches to the microhomologies flanking homothallic switching (HO) endonuclease-induced breaks and assessed their effect on MMEJ frequency and the types of repair product formation. We found that microhomology of eight to 20 base pairs carrying no more than 20% mismatches efficiently induced MMEJ. Deletion of *MSH6* did not impact MMEJ frequency. MMEJ preferentially chose a microhomology pair that was more proximal from the break. Interestingly, MMEJ events preferentially retained the centromere proximal side of the HO break, while the sequences proximal to the telomere were frequently deleted. The asymmetry in the deletional profile among MMEJ products was reduced when HO was induced on the circular chromosome. The results provide insight into how cells search and select microhomologies for MMEJ in budding yeast.

## 1. Introduction

DNA double strand breaks (DSBs) are a lethal lesion type that can lead to catastrophic cellular outcomes if not repaired quickly and accurately. Multiple mechanisms were evolved to remove DNA DSBs. Two of the most well-known DSB repair mechanisms are homologous recombination (HR) and classical non-homologous end joining (C-NHEJ), which repair DNA breaks by copying from an undamaged template across the lesion or re-ligation of broken ends after juxtaposition of broken ends together, respectively [1,2]. The third, poorly defined DSB repair mechanisms, called microhomology-mediated end joining (MMEJ) or alternative end joining (A-EJ), remove DNA breaks via annealing of microhomology flanking DSB, yielding repair products with intervening sequence deletion between microhomology pairs [3,4,5,6]. 

Because of the high mutation burden and chromosome structure changes that accompany MMEJ, MMEJ should be tightly regulated in order to serve as an alternative repair option when C-NHEJ or HR are deficient. Nevertheless, MMEJ or A-EJ operate universally in eukaryotic cells and induce a range of DNA aberrations, including telomere fusions and chromosomal rearrangements [7,8,9,10]. In yeast, DNA DSBs with no complementary overhangs produce repair events with junctional microhomology (5–15 bps) and the size of deletion ranging from a few to kilobases of nucleotides [11,12]. Oligonucleotide-mediated joining of non-complementary DNA ends and circularization of extra-chromosomal *URA4* fragments in fission yeast are also mediated by microhomology flanking DNA ends and are Ku and Lig4-independent [13]. In mammals, repair junctions containing microhomology (albeit shorter than those found in yeast) were found in Ku or Lig4 mutants during the repair of enzyme-induced break or B-and T-cell receptor gene rearrangements [14,15]. The repair of P-element excised breaks in flies and telomere fusions in plants also occurs in Lig4-independent with the repair products featuring substantial microhomologies at the junctions [16,17,18]. Evidence also suggests that the repair of Cas9 or Talen induced DNA DSBs could be mediated by flanking microhomology with the deletion of intervening sequences [19].

Biochemically, MMEJ depends on a set of proteins, many of which participate in other repair pathways, such as base excision repair or HR [3,5,6,20]. In yeast, the mechanistic similarity between NHEJ, single strand annealing (SSA), and MMEJ helps reveal a few potential biochemical steps in MMEJ [12]. According to this model, DSB is first processed to generate 3′ single stranded DNA by Mre11 and Sae2-dependent nuclease activity and to reveal embedded microhomology flanking DSB. Microhomology is then brought together by an unidentified manner independent of *RAD52* or Rad59 [10,12]. Heterotrimeric Replication Protein A (RPA) complex inhibits microhomology annealing and MMEJ [21,22]. Any non-homologous tails formed during annealing of microhomology are then cleaved by Rad1/Rad10 endonuclease, and the remaining gap is filled in by a collection of enzymes including Pol3 and Pol4 [11,12]. The nick is then ligated by Dnl4 and Cdc9-dependent manner [11]. In mammals, MMEJ depends on Polθ and LigIII for repair synthesis and ligation, respectively [8,23]. MMEJ also depends on PARP1 and XRCC3 [24]. 

MMEJ is an emerging drug target due to its role as the back-up DSB repair pathway in C-NHEJ- or HR-deficient cancers [25,26,27,28]. Polθ is highly expressed in ovarian cancer cells, and inactivation of Polθ is synthetic to *BRCA2* deficiency. Recently, inactivation of *FEN1* caused *BRCA2* synthetic growth defect due likely to its role in MMEJ [29]. Systematic and comprehensive analyses of the biochemical and the genetic components of MMEJ in multiple models are thus warranted and would facilitate the development of rationalized treatment strategies for wide ranges of cancers. Deciphering the rules and parameters of microhomology engagement and annealing should help decipher MMEJ mechanisms and allow us to predict the types of repair junctions in events associated with DNA damage and assessment of precise contributions of MMEJ to chromosomal instability in many biological events.

One of the key steps in MMEJ is microhomology annealing; yet, how cells select certain microhomology over others is unknown to date. MMEJ in yeast occurs at a lower frequency than in mammalian cells and requires higher stringency for stable pairing between paired microhomologies. The basis for this stringent microhomology annealing in budding yeast is unknown to date. 

In this study, we systematically analyzed the features of microhomology in model MMEJ reporters by altering the position, the length, and the degree of homology in microhomology. We then measured MMEJ frequency and types of MMEJ products using genetic approaches. The results revealed insights into how cells select certain microhomology for MMEJ and key parameters in dictating the types of MMEJ products in yeast.

## 2. Material and Methods

### 2.1. Yeast Strains and Plasmids

All strains were derivatives of SLY19 (*ho*Δ *MATα∷URA3∷HOcs hml*Δ*∷ADE1 hmr*Δ*∷ADE1 ade1-100 leu2-3,112 lys5 trp1∷hisG ura3-52 ade3∷GAL∷HO*)(Table 1) [30]. To construct the MMEJ reporter with different lengths and sequences of microhomology pairs, the endogenous *MAT* locus was replaced with *MAT*::*URA3*::*HO* cut site fragment of pSL19 [12] with the desired modifications introduced by different oligonucleotide pairs for PCR amplifications. The gene deletion mutants were generated by the PCR-derived *KANMX* module. The yeast strain (R072) with a circular chromosome III was a gift from Dr. James Haber [31], and the *MAT* locus was replaced with the *MAT::HO::URA3* fragment from pSL19 using the one step gene replacement technique. 

### 2.2. Homothallic Switching Endonuclease Induction

Cells were grown in pre-induction YEP-glycerol media for overnight at 30°C, and serial dilutions of cells grown to mid-log phase were plated onto YEP-agar medium containing either galactose (YEP–GAL), which induces galactose inducible homothallic switching endonuclease (HO) expression to generate DSBs, or glucose (YEPD). Survival frequency was calculated by dividing the number of colonies growing on YEP–GAL by the number of colonies growing on YEPD.

### 2.3. Analysis of Repair Events

Colonies growing on YEP–GAL plates were replica plated onto synthetic complete (SC) medium lacking uracil, and the mating type was determined by complementation test using tester strains. Uracil auxotrophs that were **a** mating type were selected for analyzing the repair events by amplifying the region spanning the repair junctions with PCR using a set of primers—pX (5′-GTAAACGGTGTCCTCTGTAAGG-3′) and p2 (5′-TCGAAAGATAAACAACCTCC-3’)—and then subjected to sequencing [11]. To examine deletional bias of DNA ends in MMEJ products, PCR of the repair products was performed using primer P2 and pI (5′-CACTCTACAAAACCAAAACCAGGG).

## 3. Results

### 3.1. Microhomology is not Randomly Selected for MMEJ

Enzymatic cleavage at two inversely oriented HO cut sites separated by ~2.0 kb *URA3* at the *MAT* locus of yeast chromosome III generated four base 3′ DNA overhangs with no complementary end sequences and whose repair relied heavily (>80%) on Ku- and *RAD52*-independent as well as Rad1-dependent MMEJ (SLY19, Figure 1A,B) [11,12]. Less than 20% of the repair still depended on Ku-dependent imprecise NHEJ (Figure 1B). Analysis of repair products revealed that the majority of the repair events (34/46 sequenced, see Table 1 in Ma et al., 2003) involved annealing of a microhomology pair composed of 12-nucleotide sequences with two mismatch bases, each situated at 2-bp centromere proximal and 60-bp distal from the breaks. Puzzlingly, in silico analysis for the available microhomologies at sequences flanking DNA break spanning several kilobases of sequences identified over three hundred microhomology pairs that were energetically as favorable as the one used predominantly in our assay (Appendix A). Furthermore, no essential genes were present at the regions spanning 7 kb and 0.5 kb from either side of the break to prevent the usage of other microhomology pairs for MMEJ. Therefore, the rarity of MMEJ events involving other microhomology pairs could not simply be explained by the lack of other microhomology at sequences flanking an HO induced DSB.

To further define how a cell selects certain microhomology for MMEJ, we deleted 12-bp microhomology proximal to the telomeric side in SLY19 and forced cells to choose other available microhomology pairs for MMEJ (Figure 1B). Surprisingly, the deletion of 12-bp microhomology, despite it being energetically less stable than others, almost completely eliminated MMEJ events (the events occurring in Ku70 deficient cells), leaving the remaining repair dependent on Ku-dependent NHEJ only (the events in *RAD1* deleted cells) (Figure 1B). Other microhomologies identified based on their energy levels and the sizes (over 12-bp) by in silico analysis failed to substitute deleted microhomology for MMEJ. The results reinforce the view that cells favor certain microhomology over others for MMEJ and underscore the importance of defining the rules and the parameters for microhomology selection.

### 3.2. MMEJ Requires Microhomology of 8- to 20-Nucleotide Long

As the first step toward defining the microhomology selection rules in MMEJ, we determined the optimum size of microhomology for MMEJ by substituting the 12-nucleotide imperfect microhomology to the 6–25 nucleotide of perfect microhomology in SLY19 and measured the frequency of MMEJ after induction of an HO DSB. The types of repair events from the survivors were analyzed by PCR amplification and sequencing of the region spanning the repair junctions using a set of oligonucleotides that annealed the 5′ to the 117-bp *MAT***a** cleavage site and the 3′ to the *MAT*α HO recognition site (Figure 1A). Strains deleted for *YKU70, RAD1,* or *RAD52* gene were analyzed in parallel to deduce the type of repair mechanisms. 

We discovered that microhomology shorter than 8-bp did not produce MMEJ at a detectable level, and the repair occurred almost exclusively by Ku-dependent imprecise NHEJ only (Figure 2). In contrast, the repair of DNA break flanked by microhomology longer than 20-bp became partially dependent on *RAD52* and corresponded to SSA. Between 8–20 nucleotide-microhomology, MMEJ frequency increased linearly as the size of microhomology increased. Ku-dependent C-NHEJ events remained constant in all strains regardless of the sizes of microhomology (Figure 2). We concluded that an efficient MMEJ requires 8- to 20-bp microhomology in yeast. We previously reached a similar conclusion from the MMEJ reporter with complementary 3′ ends [10], indicating that the size preference is universal to MMEJ in yeast. 

### 3.3. Effect of Distance between Microhomology and DNA Break on Microhomology Usage

Next, we tested if the proximity from the break dictates the frequency of microhomology selection and annealing in MMEJ. To test this, we deleted the 12-bp microhomology sequence at 60-bp distal from the break in SLY19 and relocated it to either 260-bp or 1.4-kb away from the break (Figure 3). We then measured the MMEJ frequency after HO expression in different repair pathway gene mutants. We found that relocating the 12-bp microhomology to 1.4-kb away from the break reduced MMEJ frequency (11.4%) compared to that when microhomology was located at an original location (78.2%) or at 260-bp distal to the break (67.5%), suggesting that the distance from the break represents a key parameter in dictating the frequency of MMEJ. 

To further analyze the effect of microhomology position on microhomology selection and MMEJ frequency, we generated yeast strains carrying two identical microhomologies that were telomere proximal to the break; one located at 60-bp (the original location) and the other at 260- or 1400-bp distal from the break site (Figure 3). We then examined if the distally located microhomology could successfully compete to more proximally located microhomology for MMEJ. When cells were presented with two competing microhomologies for MMEJ, proximal microhomology was used more frequently (75.2% or 80%), leaving the distal microhomology usage to only 0–6.6% among all repair events (Figure 3). 

The preferential use of proximal microhomology could be explained by the end resection that exposed the proximal microhomology prior to the distally located one [2]. To test this idea, we placed two identical microhomologies that were only 20-bp apart at the telomere proximal side of the break and assessed their usage for MMEJ after HO expression. According to the current estimation of resection rate (4 kb/h) [32], there should have been less than a 20 second difference in resecting proximal and distal microhomologies. We predicted that if the end resection dictated the proximity effect of microhomology selection in MMEJ, the longer the distance between competing microhomologies would lead to more bias in microhomology selection for MMEJ. However, even when two microhomologies were only 20-bp apart, cells still retained the proximity bias (71.4%) similar to those between microhomologies 200-bp or 1.3-kb apart. The results suggest that cells preferentially use proximal microhomology for MMEJ repair. 

### 3.4. Effect of Mismatches on Microhomology Selection

The 12-bp microhomology used overwhelmingly in our assay contained two mismatch bases (Figure 1A). We tested if the presence of mismatches within microhomology influenced the microhomology selection in MMEJ or dictated the types of repair events. To test this question, we replaced the telomere proximal 12-bp microhomology to the 20-bp microhomology with 1-, 2-, 3-, and 5-bp mismatches at various positions in SLY19 (Figure 4A). We then measured repair frequency upon induction of HO breaks. The 20-bp microhomology helped detect more broad ranges of MMEJ deficiencies. The longer microhomology also allowed additional room for variations in the position and the number of mismatches within the preferred microhomology in our assay. 

The incorporation of one to three mismatched nucleotides gradually decreased MMEJ frequency such that three base mismatches significantly (almost five-fold) reduced MMEJ frequency (Figure 4A). The five mismatches completely disabled MMEJ, leaving all repair events Ku-dependent imprecise NHEJ. Interestingly, the positions of mismatches also affected the MMEJ frequency; when two consecutive mismatches were present, the mismatches essentially reduced the size of microhomology, and the frequency of MMEJ was identical to that of the longer remaining microhomology between the two (for instance, two GG mismatches at positions 15 and 16 rendered cells to repair HO break at a frequency similar to that of 15-bp microhomology). 

To test if the inhibitory effect of mismatches on MMEJ was due to the action of the mismatch repair mechanism, yeast cells deleted for *MSH6* were constructed, and MMEJ frequency with two to five mismatched microhomology was determined. Msh2-Msh6 catalyzed heteroduplex rejection to limit homeologous recombination and correct base pair mismatches [33]. We found that deletion of *MSH6* did not alter MMEJ frequency when cells were provided with the mismatched microhomology pairs (Figure 4A). The results suggest that mismatch repair does not play detectable roles in microhomology selection and MMEJ repair. 

### 3.5. Biased Mismatch Correction in Imperfect Microhomology-Mediated MMEJ 

The mismatches within microhomology could generate two different types of MMEJ products bearing junctional sequence identical to telomere proximal or distal microhomology (Figure 4B). We analyzed the MMEJ products using two primers flanking microhomology pair with three mismatches and subjected them to sequence analysis. Surprisingly, distinct bias existed on the types of MMEJ products—among MMEJ events with three nucleotide mismatches, the final MMEJ products preferentially (90.6%) kept the full nucleotide sequence of the centromeric proximal microhomology, and even the remaining 10% carried two of the three mismatches identical to centromere proximal microhomology (Figure 4B). 

### 3.6. MMEJ Produces Repair Products Preferentially Deleting the Telomere Proximal Side of the Break

The correction bias to retain centromere proximal side of the microhomology prompted us to test if the position of microhomology on either side of the break (centromeric or telomeric side) affected microhomology usage or the types of MMEJ products. To test this idea, yeast strains (SLY19 derivatives) were constructed to carry 12- or 17- bp microhomology pairs that were located at the same distance from the break but caused the deletion of different sides of the break by MMEJ (Figure 5). We then measured the MMEJ frequency and type of the repair products after HO induction and sequence analysis. 

Surprisingly, cells almost exclusively used microhomology pairs to minimize the deletion at the centromeric side of the break for MMEJ (Figure 5). When yeast strains were induced to MMEJ using 12- or 17-bp microhomology that deleted 60-bp to the centromeric side of the break, MMEJ was dramatically reduced. Even when we placed two (12- and 17-bp) microhomology pairs flanking the break with the longer microhomology (17-bp) positioned to lead centromeric side deletion, cells rarely (6.6%) used 17-bp microhomology in MMEJ (Figure 5). The results suggest that cellular selection of microhomology pair for MMEJ depends on the location of microhomology.

### 3.7. Chromosome Circularization Disrupts Microhomology Selection Bias in MMEJ

We considered if the observed bias in MMEJ product formation reflected unique chromatin landscapes of the chromosome III. To test this possibility, we relocated our MMEJ reporter with two competing 12-bp microhomology pairs at each arm of chromosome VII (next to *SRM1* or *MUP1*) in two different orientations (Figure 5). We found that the bias in MMEJ product type persisted even at new locations, and MMEJ associated deletion was preferential to the telomeric side of the DSB end (Figure 5). The results suggest that the asymmetric deletion and the biased MMEJ product type formation is not unique to chromosome III.

We then surmised if the selection of microhomology for MMEJ was influenced by the location of the nearby telomere sequences. To test this idea, we constructed yeast strain carrying our MMEJ reporter with two microhomology pairs at the *MAT* locus of a circular chromosome III. Circular chromosome III was generated by the fusion of *HML* and *HMR* and completely lacked a telomeric sequence (Figure 5) [31]. We then measured MMEJ frequency and analyzed the type of repair products by sequencing. Surprisingly, we no longer detected microhomology selection bias (44.7 versus 55.3%) from MMEJ events in the circular chromosome, even though the overall MMEJ frequency was reduced to 75% of the linear chromosome (Figure 5).

## 4. Discussion

Microhomology annealing is central to MMEJ events and dictates the frequency and types of MMEJ products. MMEJ mysteriously uses certain microhomology sequences over others, but we do not know how cells choose microhomologies or what dictates this decision process. The lack of such information prevents us from making accurate predictions of the repair junction types in events associated with DNA damage and assessment of precise contributions of MMEJ to chromosomal instability in many biological events. In this report, we partially filled this gap of knowledge using the MMEJ reporter that almost exclusively uses 12-bp imperfect microhomology for MMEJ [12]. Our results show that the distance and the size of the microhomologies are two key parameters of microhomology selection in MMEJ.

In our experiments, MMEJ preferentially used proximal 8- to 20-bp microhomology. The repair events with longer microhomology (> 20 bp) became partially *RAD52*-dependent and thus could be mediated by SSA. We previously reported a similar size requirement on microhomology for MMEJ using a different MMEJ reporter bearing a single DSB [10]. The precise reason for >20 bp homology requirement of SSA is unknown; it could be related to the heptameric structure of *RAD52*, as each subunit binds ssDNA that is four nucleotides long [34,35]. *RAD52* might not be able to bind to microhomology less than 20-bp for SSA. Similarly, we described the effect of distance between microhomology and DNA break on MMEJ before [10], but used the assay system without competing microhomology pairs. Our study thus provides the first evidence that cells preferentially use closer microhomology for MMEJ, even when two competing microhomologies are only 20-bp apart. The apparent preference on proximal microhomology for MMEJ can be explained by their ordered availability in annealing, as the end resection proceeds linearly from the break to the distal side. However, in this scenario, two microhomologies that are 20-bp apart should be resected almost simultaneously, and the distance effect should be less apparent than when microhomologies are 200-bp or 1.3-kb apart. Alternatively, the proximity preference of microhomology selection can be attributed to the size of the 3′ flap, as the MMEJ events using distally located microhomology entail longer 3′ flaps. A great deal of evidence indicates that the size of 3′ flaps is the key determinant for efficient break-induced replication and inter-chromosomal MMEJ [36,37]. We also demonstrated that deletion of *RAD1/RAD10*, the key 3′ flap nuclease, abrogates MMEJ in several MMEJ reporters [10,11,12]. We can envision that long 3′ flaps might destabilize annealing of short microhomology. Indeed, Pol32 is specifically required for MMEJ involving longer 3′ flaps and shorter microhomology [10].

We previously showed that MMEJ frequency follows first order enzyme kinetics and depends on the melting temperature of available microhomologies [38]. Accordingly, any mismatches in microhomologies reduce the melting temperature of microhomology and MMEJ frequency [10,38]. Overall, MMEJ frequency corresponds well to the melting temperature of the given microhomologies, even though the location of mismatches also affects MMEJ frequency. The effect of mismatch positions on MMEJ is unknown to date. We discovered that Msh6 does not impact on the frequency of MMEJ using imperfect microhomologies. Interestingly, mismatches in microhomology are corrected with puzzling bias to one type over the other [37]. Although we do not know the molecular basis of this bias, the same process might control the retention of the centromeric side of the break over the telomeric side and dictate the microhomology selection and the repair types.

How does a cell preferentially retain one end during MMEJ? Evidence suggests that repair proteins are asymmetrically recruited to two DNA ends during DSB repair [39]. Association of Rad51 initiates close to the break but shows asymmetric binding pattern flanking the DSB [39]. H2A.Z distribution is also asymmetrical at the centromeric and the telomeric side of the break [39]. Notably, we reported that deletion of *RAD52* did not affect the MMEJ frequency but altered the types of MMEJ products [11]. It should be possible that asymmetric association of *RAD52* or other repair factors at DSB might modulate microhomology selection/annealing and the types of repair products. We also propose that repair protein distribution is altered in circular chromosome to attenuate the deletional bias during MMEJ. Additional experiments are necessary to identify factors responsible for end retention bias in MMEJ and their distributions in linear and circular chromosomes.

## 5. Conclusions

Taken together, we propose that selection of microhomology for MMEJ is non-random and likely follows certain rules and parameters to maximize repair efficiency and fidelity. The size, the position, and the mismatches of microhomology in aggregate dictate the repair outcome in conjunction with a not yet identified mechanism to retain one end of the break for MMEJ. Elucidation of the underlying mechanism of directional and positional bias in microhomology selection is likely important and would shed light on MMEJ-mediated chromosomal aberrations and mutagenesis.

## Figures and Tables

**Figure 1 genes-10-00284-f001:**
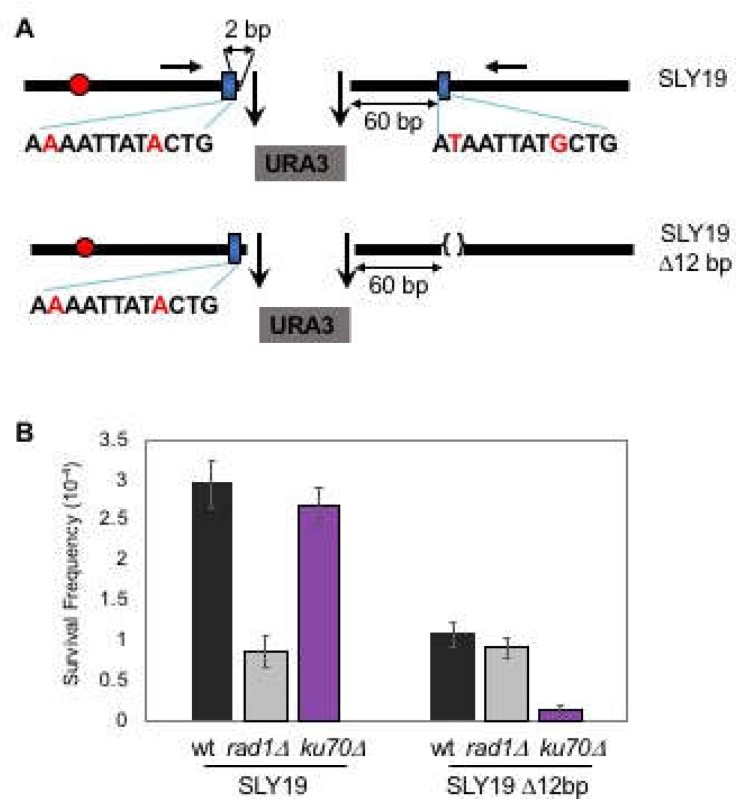
Microhomology selection for microhomology-mediated end joining (MMEJ) is non-random. (**A**) Two homothallic switching endonuclease (HO) cleavages at the *MAT* locus (the 117-bp *MAT***a** and the full-length *MAT*α cut sites) separated by ∼2 kb of *URA3* sequence preferentially induced Ku and *RAD52*-independent Microhomology-mediated end joining (MMEJ) using the flanking 12-bp of imperfect microhomology that was 2-bp and 60-bp away from the break. The location of the centromere (red circles) and the preferred microhomology (blue boxes) for MMEJ are shown. The sequence (mismatches are shown in red characters) of the preferred microhomology are shown below the blue boxes. The positions of the primers for PCR amplification and sequencing are shown by red arrows. (**B**) Survival frequency after induction of HO breaks of the wild type (WT), the *RAD1* gene deleted [*RAD1*Δ representing the frequency of classical non-homologous end joining (C-NHEJ)], and the *KU70* deletion derivatives (*ku70*Δ, the frequency of MMEJ) of SLY19 or SLY19 lacking preferred 12-bp imperfect microhomology (SLY19 Δ12-bp). The frequency of survival after an HO-induced double strand breaks (DSB) was calculated by dividing the number of colonies growing on YEP-agar medium containing galactose (YEP–GAL) by the number of colonies growing on YEP-agar medium containing glucose (YEPD). Each value represents the average from at least three independent experiments ± standard deviation.

**Figure 2 genes-10-00284-f002:**
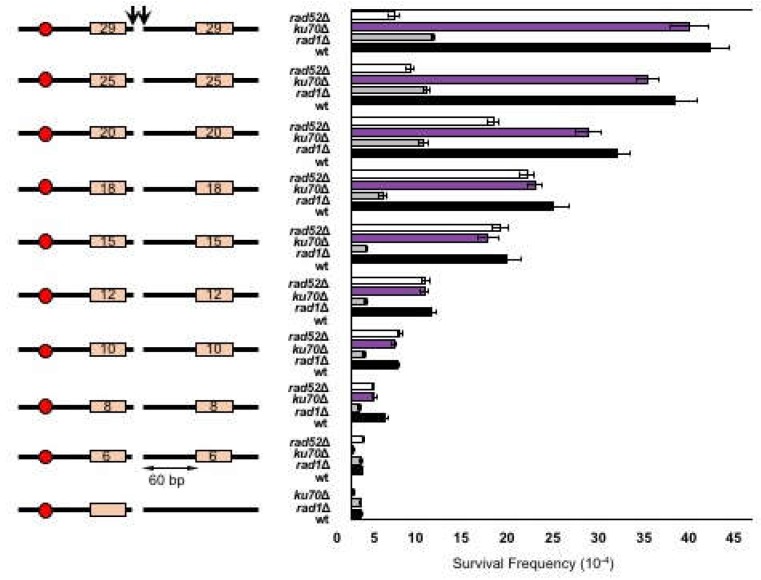
MMEJ requires 8-20 bp microhomology. (**Left**) A diagram of MMEJ reporters with different sizes of microhomology flanking an HO break. The locations of HO cut site (arrow), microhomology (orange boxes), and centromere (red circle) are shown. The size of microhomology is shown in number (bp) inside the microhomology. (**Right**) Graph showing survival frequency ± standard deviation (SD) in wild type (WT), *RAD1*Δ (MMEJ deficient mutant), *ku70*Δ (C-NHEJ deficient mutant), and *RAD52*Δ [homologous recombination (HR) deficient mutant]. Survival frequency was calculated by dividing the number of colonies surviving on the YEP–GAL plates by the number of colonies surviving on the YEPD plates. The results are the average of three independent experiments.

**Figure 3 genes-10-00284-f003:**
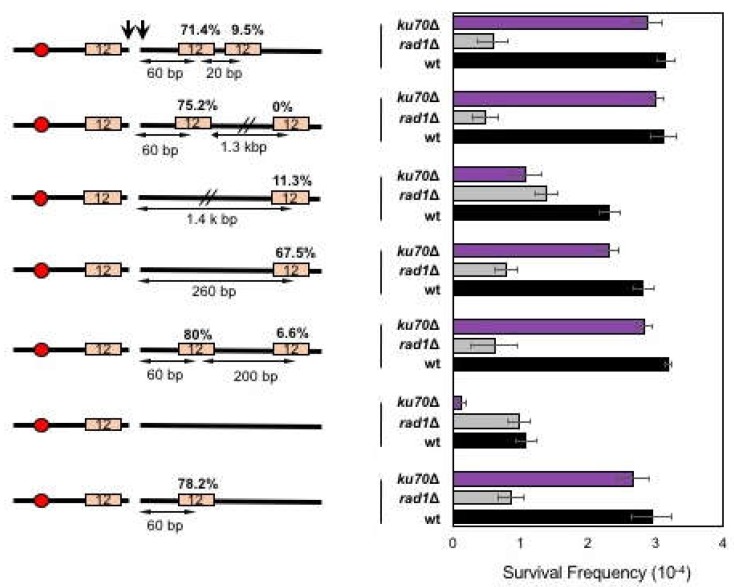
The effect of proximity on microhomology selection for MMEJ. (**Left**) A diagram of MMEJ reporters with microhomologies located at different distances to an HO break. The locations of HO cut site (arrow), microhomology (orange boxes), and centromere (red circles) are shown. The size of microhomology is shown in number (bp) inside the microhomology. The distance to the break (bp) is shown below each microhomology. Percentage of repair event was calculated by dividing the number of repair events using each microhomology with the number of all repair events, regardless of the repair types and shown above each microhomology. The repair events were analyzed by sequencing of the repair junctions from >100 Ura- survivors. (**Right**) Graph showing survival frequency ± SD in wild type (WT), *RAD1*Δ (MMEJ deficient mutant), and *ku70*Δ (C-NHEJ deficient mutant). Survival frequency was calculated as described in Figure 1. The results are the average of three independent experiments.

**Figure 4 genes-10-00284-f004:**
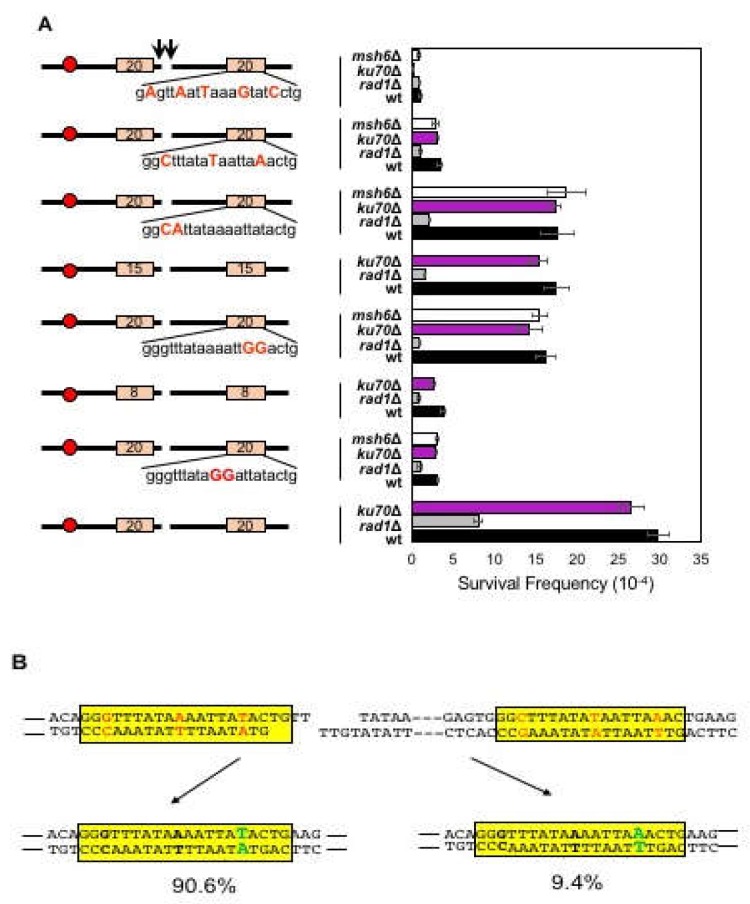
The effect of mismatches on microhomology selection for MMEJ. (**A**) Left: A diagram of MMEJ reporters with imperfect microhomologies carrying different numbers and positions of mismatches. The locations of HO cut site (arrow), microhomology (orange boxes), and centromere (red circles) are shown. The size of microhomology is shown in number (bp) inside the microhomology. The sequence of microhomology and the position of mismatch (highlighted in red) are shown. Right: Graph showing survival frequency ± s.d. in wild type (WT), *RAD1*Δ (MMEJ deficient mutant), *ku70*Δ (C-NHEJ deficient mutant), and *msh6*Δ (mismatch repair deficient mutant). Survival frequency was calculated as described in Figure 1. The results are the average of three independent experiments. (**B**) Two different ways (highlighted in green characters) mismatches (shown in red) in microhomology were repaired. The corrected bases are shown in bold characters. The microhomology is shown in yellow boxes. Analyses of 94 MMEJ products showed strong bias to one type over the other. The percentage of repair product type among total MMEJ repair events was calculated based on sequencing of the repair junctions from the Ura- survivors after HO expression.

**Figure 5 genes-10-00284-f005:**
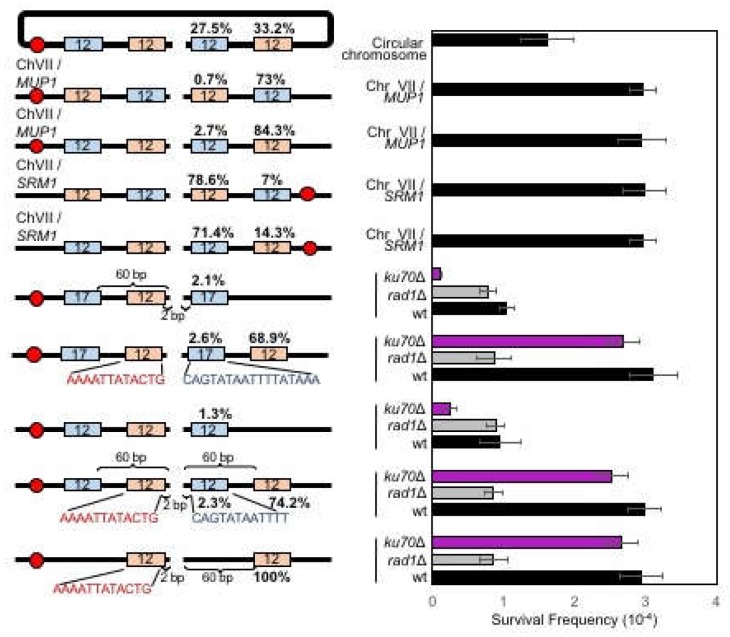
The position effect of microhomology on MMEJ. Left: A diagram of MMEJ reporters with microhomology pairs at different locations to the HO break. The locations of HO cut site (arrow), microhomology (orange and blue boxes), and centromere (red circles) are shown. The size of microhomology is shown in number (bp) inside the microhomology. The distance to the break (bp) is shown above each microhomology. The size of the 3′ flap (bp) and the sequences of microhomologies are also shown. Percentage of repair event was calculated by dividing the number of repair events using each microhomology with the number of all repair events, regardless of the repair types and shown above each microhomology. The repair events were analyzed by sequencing of the repair junctions from >100 Ura- survivors. Right: Graph showing survival frequency ± s.d. in wild type (WT), *RAD1*Δ (MMEJ deficient mutant), and *ku70*Δ (C-NHEJ deficient mutant). Survival frequency was calculated as described in Figure 1. The results are the average of three independent experiments.

**Table 1 genes-10-00284-t001:** Strain List.

SLY19	*ho*Δ *MATα∷URA3∷HOcs hml*Δ*∷ADE1 hmr*Δ*∷ADE1 ade1-100 leu2-3,112 lys5 trp1∷hisG ura3-52 ade3∷GAL∷HO*
YKHL76	SLY19 *ku70*Δ*::KAN*
YKHL77	SLY19 *RAD1*Δ*::KAN*
YKHL101	SLY19 *12bp microhomology (MH)*Δ
YKHL102	JKM179 *MAT::URA3::HO::6MH*
YKHL103	YKHL102 *ku70*Δ*::KAN*
YKHL104	YKHL102 *RAD1*Δ*::KAN*
YKHL105	JKM179 *MAT::URA3::HO::10MH*
YKHL106	YKHL105 *ku70*Δ*::KAN*
YKHL107	YKHL105 *RAD1*Δ*::KAN*
YKHL108	JKM179 *MAT::URA3::HO::12MH*
YKHL109	YKHL108 *ku70*Δ*::KAN*
YKHL111	YKHL108 *RAD1*Δ*::KAN*
YKHL112	JKM179 *MAT::URA3::HO::15MH*
YKHL113	YKHL112 *ku70*Δ*::KAN*
YKHL114	YKHL112 *RAD1*Δ*::KAN*
YKHL115	JKM179 *MAT::URA3::HO::18MH*
YKHL116	YKHL115 *ku70*Δ*::KAN*
YKHL117	YKHL115 *RAD1*Δ*::KAN*
YKHL118	JKM179 *MAT::URA3::HO::20MH*
YKHL119	YKHL118 *ku70*Δ*::KAN*
YKHL121	YKHL118 *RAD1*Δ*::KAN*
YKHL122	YKHL118 *rad52*Δ*::KAN*
YKHL123	JKM179 *MAT::URA3::HO::25MH*
YKHL124	YKHL123 *ku70*Δ*::KAN*
YKHL125	YKHL123 r*ad1*Δ*::KAN*
YKHL126	YKHL123 *rad52*Δ*::KAN*
YKHL127	JKM179 *MAT::URA3::HO::29MH*
YKHL128	YKHL127 *ku70*Δ*::KAN*
YKHL129	YKHL127 *RAD1*Δ*::KAN*
YKHL131	YKHL127 *rad52*Δ*::KAN*
YKHL132	JKM179 *MAT::URA3::HO::200bp::12MH*
YKHL133	YKHL132 *ku70*Δ*::KAN*
YKHL134	YKHL132 *RAD1*Δ*::KAN*
YKHL142	JKM179 *MAT::URA3::HO::12MH::200bp::12MH*
YKHL143	YKHL142 *ku70*Δ*::KAN*
YKHL144	YKHL142 *RAD1*Δ*::KAN*
YKHL145	JKM179 *MAT::URA3::HO::12MH::20bp::12MH*
YKHL146	YKHL145 *ku70*Δ*::KAN*
YKHL147	YKHL145 *RAD1*Δ*::KAN*
YKHL148	JKM179 *MAT::URA3::HO::12MH w/ mismatch gggtttataGGattatactg*
YKHL149	YKHL148 *ku70*Δ*::KAN*
YKHL151	YKHL148 *RAD1*Δ*::KAN*
YJHJ1	YKHL148 *msh6*Δ*::KAN*
YKHL152	JKM179 *MAT::URA3::HO::12MH w/ mismatch gggtttataaaattGGactg*
YKHL153	YKHL152 *ku70*Δ*::KAN*
YKHL154	YKHL152 *RAD1*Δ*::KAN*
YJHJ2	YKHL152 *msh6*Δ*::KAN*
YKHL155	JKM179 *MAT::URA3::HO::12MH w/ mismatch ggCAttataaaattatactg*
YKHL156	YKHL155 *ku70*Δ*::KAN*
YKHL157	YKHL155 *RAD1*Δ*::KAN*
YJHJ3	YKHL155 *msh6*Δ*::KAN*
YKHL158	JKM179 *MAT::URA3::HO::12MH w/ mismatch ggCtttataTaattaAactg*
YKHL159	YKHL158 *ku70*Δ*::KAN*
YKHL161	YKHL158 *RAD1*Δ*::KAN*
YJHJ4	YKHL158 *msh6*Δ*::KAN*
YKHL165	JKM179 *MAT::URA3::HO::12MH w/ mismatch gAgttAatTaaaGtatCctg*
YKHL166	YKHL165 *ku70*Δ*::KAN*
YKHL167	YKHL165 *RAD1*Δ*::KAN*
YKHL179	YKHL152 *msh6*Δ*::KAN*
YKHL182	JKM179 *MAT::URA3::12MH-1::12MH-2::HO::12MH-1::12MH-2*
YKHL183	YKHL182 *ku70*Δ*::KAN*
YKHL184	YKHL182 *RAD1*Δ*::KAN*
YKHL185	JKM179 *MAT::URA3::12MH::12MH::HO::12MH*
YKHL186	YKHL185 *ku70*Δ*::KAN*
YKHL187	YKHL185 *RAD1*Δ*::KAN*
YKHL188	JKM179 *MAT::URA3::17MH::12MH::HO::17MH::12MH*
YKHL189	YKHL188 *ku70*Δ*::KAN*
YKHL201	YKHL188 *RAD1*Δ*::KAN*
YKHL202	JKM179 *MAT::URA3::17MH::12MH::HO::17MH::12MH*
YKHL203	YKHL188 *ku70*Δ*::KAN*
YKHL204	YKHL188 *RAD1*Δ*::KAN*
YKHL208	JKM179 *MAT*Δ *Mup1::12MH-2::12MH-1::HO::12MH-2::12MH-1*
YKHL209	JKM179 *MAT*Δ *Mup1::12MH-1::12MH-2::HO::12MH-1::12MH-2*
R072	described at Haber and Thorburn (1984)
YKHL211	R072 circular ChIII *MAT::URA3::HO::12MH::20bp::12MH*

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
