# Peer review of "Microhomology Selection for Microhomology Mediated End Joining in *Saccharomyces cerevisiae"

_genes, 2019, doi:10.3390/genes10040284_

Round 1
Reviewer 1 Report
In this manuscript, Lee et al. applied genetic approaches to investigate the factors that affect microhomology (MH) selection during MMEJ process. They varied the length, the relative position, and the degree of homology (0 – 5 mismatches in total 20 bp) in the microhomologies flanking HO-induced DSB in model MMEJ reporters, and found that
1. MMEJ requires a microhomology of 8- to 20-nucleotide long;
2. More than 20% mismatches (3 mismatches / 20 bp MH) within MH severely reduces the MMEJ frequency;
3. MMEJ prefers to proximal microhomology from the break;
4. MMEJ usually retains the sequences of centromere proximal microhomology of the DSB, while the sequences proximal to the telomere side are frequently deleted.
The data presented here provide important insights into microhomology selection rules of MMEJ repair in yeast.
1. Line 118, the authors demonstrated that majority of repair events involve the annealing of the 12 nucleotide microhomology. Do the authors have the accurate ratio of the repair events using the 12-nucleotide microhomology to the total sequenced events? How did the authors sequence the repair junctions? As shown in Fig. 1A, the primers designed for junction sequencing are specific for the most proximal microhomologies flanking the DSB site. How could the sequencing cover the repair junctions from the usage of distal microhomologies (hundred- and thousand-nucleotide away from the DSB site)? The authors emphasized the usage of the proximal MH here by just mentioning majority of repair events involving the annealing of the proximal 12 nucleotide microhomology. The sequencing results should be listed. We are not asking for additional sequencing, but merely a clearer explanation of what was done.
2. What do the ratios on Fig.3 left, Fig.4B, and Fig. 5 left mean? They seem representing different concepts. The details about the ratio calculation should be shown in the legends. Moreover, how many repair junctions were sequenced for the three-mismatch containing MH shown in Fig. 4B? Do the MMEJ repair also bias mismatch correction to the telomere proximal microhomology for two-mismatch containing MH?
3. How large are the HO cleavage sites in each experiment? The detailed overhang sequences should be shown on the graphs or in the legends.
4. Can the authors predict the MH selection preference if one or two mismatches are present in the proximal MH, but not in the distal MH for the competitive experiments shown in Fig.3?
5. In Fig.5, what are the survival frequencies using circular chromosome in ku70 deletion or rad 1 deletion strain? It looks like 40% of HO-cut DSB on circular chromosome were repaired through NHEJ pathway, while less than 20% DSB on linear chromosome were repaired by NHEJ pathway. Can the authors discuss the possible reasons causing the differences? The sequences of the two competitive 12-nucleotide MHs in Fig. 5 should be shown.
6. What is the possible mechanism of repair products preferentially deleting the telomere proximal side of the break? Does the polarity of the DSB overhang lead to the preference? The authors may test this possibility using blunt ends or 5’ overhang DSB ends. The polarity of the ends may also result in the biased mismatch correction within imperfect MH. Another possibility is that the differences of distance between the first essential gene and MH (either centromeric side or telomeric side) contribute to this bias.
7. In the Discussion, line 354, the authors demonstrated that “it should be possible that asymmetric association of Rad52 … might modulate microhomology selection/annealing and the types of repair products”. But MMEJ process is Rad52-independent as the authors also demonstrated in the previous parts. Can the authors explain the connections between the asymmetric binding of Rad52 to DSB and the biased mismatch correction and deletion by MMEJ repair?
Author Response
Reviewer #1.
The data presented here provide important insights into microhomology selection rules of MMEJ repair in yeast.
I am very thankful to the reviewer’s positive comments and helpful suggestions.
We addressed all of the reviewer’s comments and the manuscript is now greatly improved and should be suitable for publication in Genes.
1. Line 118, the authors demonstrated that majority of repair events involve the annealing of the 12 nucleotide microhomology. Do the authors have the accurate ratio of the repair events using the 12-nucleotide microhomology to the total sequenced events?
The accurate ratio (34 out of 46 sequenced events) of the MMEJ events using the 12 nucleotide, imperfect microhomology (MH) were described in our previously published manuscript (Ma et al., 2003). The results are reminded in the revision (Page 8, line 4).
How did the authors sequence the repair junctions? As shown in Fig. 1A, the primers designed for junction sequencing are specific for the most proximal microhomologies flanking the DSB site. How could the sequencing cover the repair junctions from the usage of distal microhomologies (hundred- and thousand-nucleotide away from the DSB site)?
We recovered the repair junctions by PCR from surviving colonies using the primer set shown in the Figure 1. We analyzed the types of repair products from >300 surviving colonies by PCR and all but 7 produced PCR products, indicating that the MMEJ with distal MHs are rare events (<3%).
The authors emphasized the usage of the proximal MH here by just mentioning majority of repair events involving the annealing of the proximal 12 nucleotide microhomology. The sequencing results should be listed. We are not asking for additional sequencing, but merely a clearer explanation of what was done.
The sequencing results were already described in Ma et al (2003). We cited the paper to refer the sequencing results.
2. What do the ratios on Fig.3 left, Fig.4B, and Fig. 5 left mean? They seem representing different concepts. The details about the ratio calculation should be shown in the legends.
The ratios on Fig. 3 and 5 indicate the percentage of repair events that use the corresponding microhomology among total repair events. In contrast, the ratios on Fig. 4B represent the percentage of repair product types among the MMEJ events that are fully sequenced. The details how we calculated each percentage are described in the figure legends.
Moreover, how many repair junctions were sequenced for the three-mismatch containing MH shown in Fig. 4B? Do the MMEJ repair also bias mismatch correction to the telomere proximal microhomology for two-mismatch containing MH?
We analyzed the sequences of 94 MMEJ repair junctions that use microhomology with three mismatches. We previously reported that microhomology with two base pair mismatches are preferentially corrected with the bias to the telomere proximal microhomology (85.2%) in MMEJ (Ma et al., 2003, see Table 1).
3. How large are the HO cleavage sites in each experiment? The detailed overhang sequences should be shown on the graphs or in the legends.
The MATa HO cleavage site is 117 bp long. We employed the full-length MATa HO cleavage site without any modification. The overhang sequence is shown in Fig. 5.
4. Can the authors predict the MH selection preference if one or two mismatches are present in the proximal MH, but not in the distal MH for the competitive experiments shown in Fig.3?
We agree with the reviewer that it will be interesting to test how different parameters and rules interact and integrate to modulate MH selection in MMEJ although it will be beyond the scope of this paper. We will be pursuing the experiments as the follow-up.
5. In Fig.5, what are the survival frequencies using circular chromosome in ku70 deletion or rad 1 deletion strain? It looks like 40% of HO-cut DSB on circular chromosome were repaired through NHEJ pathway, while less than 20% DSB on linear chromosome were repaired by NHEJ pathway. Can the authors discuss the possible reasons causing the differences? The sequences of the two competitive 12-nucleotide MHs in Fig. 5 should be shown.
We did not construct ku70 or rad1 deletion derivatives of the strain carrying a circular chromosome III. We thus do not know survival frequencies in KU or RAD1 deleted cells after HO induced DSB formation at circular chromosome. The higher percentage of NHEJ among survivors after HO induced DSB at circular chromosome flanking 12 bp microhomology can be explained simply by the lower frequency of MMEJ after HO expression. The actual NHEJ frequency according to PCR analyses of the repair junctions is not significantly different from that of linear chromosomes (0.85 vs 1.05 x 10-4, linear or circular chromosome, respectively). We do not know the reason for the low MMEJ frequency when DSB is flanked by MHs at circular chromosome.
The sequences of the two competing 12 bp microhomologies in Fig. 5 are now shown in response to the reviewer’s comments.
6. What is the possible mechanism of repair products preferentially deleting the telomere proximal side of the break?
We do not know the basis why cells preferentially delete telomeric side in MMEJ products.
Does the polarity of the DSB overhang lead to the preference? The authors may test this possibility using blunt ends or 5’ overhang DSB ends. The polarity of the ends may also result in the biased mismatch correction within imperfect MH.
It is an interesting idea if the types and the polarity of DSB ends could dictate mismatch correction and deletion side bias in MMEJ. We wish to analyze this and several other possibilities to solve this puzzle but it will be beyond the scope of this paper.
Another possibility is that the differences of distance between the first essential gene and MH (either centromeric side or telomeric side) contribute to this bias.
The distance to essential genes from DSB has no correlation to the deletional bias in MMEJ product formation. The distance to essential gene is identical between the MMEJ reporters at linear and circular chromosome whereas the deletional bias is significantly reduced only in MMEJ reporter at circular chromosome.
7. In the Discussion, line 354, the authors demonstrated that “it should be possible that asymmetric association of Rad52 … might modulate microhomology selection/annealing and the types of repair products”. But MMEJ process is Rad52-independent as the authors also demonstrated in the previous parts. Can the authors explain the connections between the asymmetric binding of Rad52 to DSB and the biased mismatch correction and deletion by MMEJ repair?
Even if MMEJ frequency is not dependent on Rad52, Rad52 likely has some effect on MMEJ process because deletion of RAD52 alters the types of MMEJ products in cells after two inversely oriented HO break formation (Ma et al., 2003). We surmise that MMEJ requires the formation of single stranded DNA (ssDNA), at which Rad52 associate by virtue of its affinity to ssDNA. We speculate if the binding of Rad52 at ssDNA might impact on one or more steps of MMEJ and thus the types of repair products.
Reviewer 2 Report
In yeast, the efficiency of microhomology-mediated end joining (MMEJ) depends on the length of the microhomology pairs and their sequence identity. Another factor that influences the MMEJ efficiency is the distance from the DSB. Villarreal et al (2012) have shown that increasing the length of the microhomologies (MH) between 12 and 18 bp increased the frequency of MMEJ, whereas increasing the number of mismatches within the 18 bp MH decreased it. Furthermore, they found that increasing the distance of the microhomologies from the double strand break (DSB) reduces MMEJ frequency, suggesting flap removal plays a rate limiting step in MMEJ. In the present study, Lee et al. aim to expand on their previous study by further testing the parameters of their previous study. They replicated some of their earlier studies in a different chromosomal context, confirming that MH length, mismatch density and distance from ends are general rules for MMEJ in yeast. The major novel, and interesting, finding in the current study is the fact that MMEJ exhibits a bias in the choice of microhomologies, favoring those that lead to less deletion on the centromeric side of the DSB.
Major comments:
1. Many of the results are quite redundant with those in in Villareal et al (2012). This includes the findings that the length of the MH, their distance from the break and the number of mismatches all affect MMEJ frequency. Figure 4a particularly lacks in the novelty aspect.
2. In figure 2, the effect of the length of the MHs on MMEJ frequency appears to be less drastic than what has been reported in Villareal et al (2012). For instance, in the present study, changing the length from 12bp to 15bp increases the frequency by only 2-fold, whereas such a change in the previous study resulted in at least 20-fold difference. Is there an explanation for this discrepancy?
3. In figure 4, the 20bp microhomology length is used to test the effect of the length of the mismatches on MMEJ frequency. However, as seen in figure 3, the repair events at this length become partially dependent on Rad52, suggesting some of the events examined in figure 4 are SSA rather than MMEJ-mediated. Why not use the 18bp microhomology length instead?
4. In the Villareal study, moving the MH further than 2bp away from the break dramatically reduces the frequency of MMEJ. In figure 5, there is no mention of the distance from the break of the MH closest to the break on the telomeric side. Are those MH also 2bp away? The depiction on the schematic seems to imply that they are further away than those on the centromeric side. This is an important point to clarify because if the length of the flap that needs to be removed is critical, and if it is shorter on the centromeric side than the telomeric side, it could explain the bias. For all the constructs shown in Fig 5 please indicate the # bp between the MH and DSB.
5. It would be helpful to also show the data for MMEJ on the other side of Chromosome VII (with microhomologies next to MUP1) (Fig 5).
6. Is the same system with the inverted HO cut site (inducing two DSBs) used in all the experiments? This should be clarified.
7. In figure 2, increasing the length of the microhomologies past 18bp appears to decrease the frequency of MMEJ, as seen in the rad52 mutants. This is a potentially interesting observation that is not addressed in the paper.
8. In reference to figure 2, the authors suggest that the C-NHEJ events remained constant across all strains. However, the frequency of the Rad1-independent events seems to increase when the length of the microhomologies is increased to more than 15bp. Can the authors speculate as to whether those events are in fact C-NHEJ events, Ku-independent events or MMEJ that is less dependent on Rad1? Perhaps the more stable pairing between longer MHs allows a normally less efficient Rad1-independent mechanism for flap removal.
Minor comments:
1. The rationale for the present study, as compared to the Villareal study, could be expanded on in the introduction.
2. I think it is important to make it clear to the reader that MMEJ in yeast occurs at much lower frequency than in mammalian cells and with much higher stringency for stable pairing between paired MHs. This is most likely due to the absence of Pol theta in yeast.
3. The paper needs editorial changes with regards to syntax and some mistakes (e.g. in page 10, line 268, in should say Fig 5 instead of Fig 5A).
4. Line 26, DSBs are induced on a circular chromosome, not HO.
Author Response
Reviewer #2.
I am very thankful to the reviewer’s positive comments and helpful suggestions.
We addressed all of the reviewer’s comments and the manuscript is now significantly improved and should be suitable for publication in Genes.
Major comments:
1. Many of the results are quite redundant with those in in Villareal et al (2012). This includes the findings that the length of the MH, their distance from the break and the number of mismatches all affect MMEJ frequency. Figure 4a particularly lacks in the novelty aspect.
Unlike the MMEJ reporter used in Villarreal et al., the current manuscript reports our study that used the two inversely oriented HO break reporter and analyzed the effect of MH length, their distance to the break, and the number of mismatches on MMEJ. The results identify the universal features of MMEJ in MH selection across different MMEJ reporters. We also analyzed the role of mismatch repair in MMEJ with imperfect MHs, which we have reported here for the first time.
2. In figure 2, the effect of the length of the MHs on MMEJ frequency appears to be less drastic than what has been reported in Villarreal et al (2012). For instance, in the present study, changing the length from 12bp to 15bp increases the frequency by only 2-fold, whereas such a change in the previous study resulted in at least 20-fold difference. Is there an explanation for this discrepancy?
We do not know why we detected less dramatic effect of the MH size on MMEJ frequency in the two inversely oriented HO break reporter. Overall, MMEJ frequency is much lower in two HO break reporter system than that with one HO break used in Villarreal et al.
3. In figure 4, the 20bp microhomology length is used to test the effect of the length of the mismatches on MMEJ frequency. However, as seen in figure 3, the repair events at this length become partially dependent on Rad52, suggesting some of the events examined in figure 4 are SSA rather than MMEJ-mediated. Why not use the 18bp microhomology length instead?
MMEJ frequency is quite low in two inversely oriented HO break reporter and the effect of one or more mismatches on MMEJ is harder to reliably detect if the size of MH is smaller in the current reporter system. We would also like to emphasize that although the repair events in cells with 20 MH is partially dependent on Rad52 (~20%), MMEJ is still the primary repair option and therefore the effect of mismatches on 20 bp MH repair should be relevant to MMEJ events.
4. In the Villarreal study, moving the MH further than 2bp away from the break dramatically reduces the frequency of MMEJ. In figure 5, there is no mention of the distance from the break of the MH closest to the break on the telomeric side. Are those MH also 2bp away? The depiction on the schematic seems to imply that they are further away than those on the centromeric side.
All the reporters in the paper employed the closest MHs that are 2 bp away from the break.
This is an important point to clarify because if the length of the flap that needs to be removed is critical, and if it is shorter on the centromeric side than the telomeric side, it could explain the bias. For all the constructs shown in Fig 5 please indicate the # bp between the MH and DSB.
We ruled out the effect of 3’ flap length on deletional bias in MMEJ by constructing MMEJ reporters where two sets of MH are located at either side of the break with the identical size (2-bp) 3’ flaps after MH annealing. These reporters still retained centromeric side MHs in MMEJ products and deleted the telomeric side, suggesting that the length of 3’ flap does not dictate deletional orientation bias in MMEJ.
We also described the distance between the break and the MHs in Fig. 5 as suggested by the reviewer.
5. It would be helpful to also show the data for MMEJ on the other side of Chromosome VII (with microhomologies next to MUP1) (Fig 5).
The revision includes the MUP1 results in Fig. 5.
6. Is the same system with the inverted HO cut site (inducing two DSBs) used in all the experiments? This should be clarified.
Yes, we clarified this point throughout the manuscript by indicating that all MMEJ strains are the derivatives of SLY19 (the strain carrying two inversely oriented HO breaks).
7. In figure 2, increasing the length of the microhomologies past 18bp appears to decrease the frequency of MMEJ, as seen in the rad52 mutants. This is a potentially interesting observation that is not addressed in the paper.
We described in the manuscript that the frequency of repair events using MHs longer than 18 bp are partially dependent on Rad52 and thus could be categorized as single strand annealing. By definition, we did not consider repair events dependent on Rad52 as non-MMEJ events.
8. In reference to figure 2, the authors suggest that the C-NHEJ events remained constant across all strains. However, the frequency of the Rad1-independent events seems to increase when the length of the microhomologies is increased to more than 15bp. Can the authors speculate as to whether those events are in fact C-NHEJ events, Ku-independent events or MMEJ that is less dependent on Rad1? Perhaps the more stable pairing between longer MHs allows a normally less efficient Rad1-independent mechanism for flap removal.
This is an excellent point, which we would like to pursue as a follow up study. It is however beyond the scope of the current paper.
Minor comments:
1. The rationale for the present study, as compared to the Villarreal study, could be expanded on in the introduction.
We expanded the introduction briefly to emphasize the rationale and the broad impact of the present study on MMEJ mechanisms as suggested by the reviewer.
2. I think it is important to make it clear to the reader that MMEJ in yeast occurs at much lower frequency than in mammalian cells and with much higher stringency for stable pairing between paired MHs. This is most likely due to the absence of Pol theta in yeast.
We wrote in the introduction that MMEJ in yeast occurs at the lower frequency and requires high stringency for stable MH pairing than in mammalian cells.
3. The paper needs editorial changes with regards to syntax and some mistakes (e.g. in page 10, line 268, in should say Fig 5 instead of Fig 5A).
We edited the manuscript to fix these and other mistakes.
4. Line 26, DSBs are induced on a circular chromosome, not HO.
We revised the paragraph to fix this error.